# Thermally-nucleated self-assembly of water and alcohol into stable structures at hydrophobic interfaces

Kislon Voïtchovsky[1], Daniele Giofrè[2], Juan José Segura[2], Francesco Stellacci[2,3] & Michele Ceriotti[2]

At the interface with solids, the mobility of liquid molecules tends to be reduced compared with bulk, often resulting in increased local order due to interactions with the surface of the solid. At room temperature, liquids such as water and methanol can form solvation structures, but the molecules remain highly mobile, thus preventing the formation of long-lived supramolecular assemblies. Here we show that mixtures of water with methanol can form a novel type of interfaces with hydrophobic solids. Combining *in situ* atomic force microscopy and multiscale molecular dynamics simulations, we identify solid-like two-dimensional interfacial structures that nucleate thermally, and are held together by an extended network of hydrogen bonds. On graphite, nucleation occurs above ~35 °C, resulting in robust, multilayered nanoscopic patterns. Our findings could have an impact on many fields where water-alcohol mixtures play an important role such as fuel cells, chemical synthesis, self-assembly, catalysis and surface treatments.

[1] Department of Physics, Durham University, Durham DH1 3LE, UK. [2] Institute of Materials, Ecole Polytechnique Fédérale de Lausanne, 1015 Lausanne, Switzerland. [3] Interfaculty Bioengineering Institute, Ecole Polytechnique Fédérale de Lausanne, Switzerland. Correspondence and requests for materials should be addressed to K.V. (email: kislon.voitchovsky@durham.ac.uk).

At the interface with solids, liquid molecules usually exhibit important differences in their behaviour when compared with bulk liquid at equilibrium. Their interaction with the solid, together with reduced configurational entropy, often results in a loss of molecular mobility[1,2], and depending on the system, local ordering[3,4]. Confinement or cooling can promote supramolecular order at the interface. Water, for example, was shown to form entirely new two-dimensional (2D) phases when sandwiched between graphene sheets[5,6] or on metal surface[1,2,7] at low temperature[3,4,8]. In ambient conditions and in the absence of confinement, water molecules tend to remain liquid at the interface with most immersed solids at room temperature. Their mobility, even if reduced, does not allow for the formation of ice-like arrangement with a stable, long-lived organized network of hydrogen bonds[5,6,9]. The same holds for short alcohols such as ethanol or methanol, whose hydrophobic moiety is too small to enable van der Waals-based interfacial self-assembly[9]. Solutions containing both water and alcohol tend to create a glassy, alcohol-rich layer at hydrophobic interfaces[10,11], but stable molecular arrangements have only been observed at low temperatures[12], and little is known about their behaviour at room temperature. Part of the difficulty comes from the fact that although water and alcohol form a stable bulk solution at all concentrations, the liquid is far from homogeneous at the molecular level[13–15].

Over the last decade, a combination of experimental[13–15] and theoretical[16,17] studies have established that both solvents tend to form segregated clusters or molecular chains with alternating water and alcohol molecules that percolate through the liquid. These structures are highly dynamical and the liquid molecules constantly rearrange, but some preferred molecular arrangements can be found such as clusters, chains or even rings, depending on the concentration of alcohol in the water[13,15–17]. Although transient, these structures consistently reduce the entropy of mixing of the solution when compared with an ideal, perfectly homogenous mixture[13]. The influence of this nanoscale order on the behaviour and molecular organization at interfaces with solids is not known.

Here we examine the molecular organization adopted by solutions of water and simple alcohols such as methanol and ethanol at the interfaces with highly ordered pyrolytic graphite (HOPG). Using *in situ* atomic force microscopy (AFM) in liquid with temperature control, we image the molecular arrangement at the interface with sub-nanometre resolution. The AFM results are complemented by density functional theory (DFT) and molecular dynamics (MD) simulations of the HOPG/water–methanol interface. The results show that water and alcohol molecules can form stable, solid-like 2D layered assemblies that include both types of molecules at the interface with immersed hydrophobic solids. The structures remain stable at room temperature, pointing to the collective effect of an extended H-bond network. This is significant because layering of long alcohol molecules at interfaces is usually based on van der Waals interactions[18,19], which are not strong enough to drive the self-assembly of short alcohols such as methanol or ethanol at the interface with immersed solids at room temperature.

## Results

**High-resolution atomic force microscopy**. Experimentally, it is possible to visualize solid–liquid interfaces locally, *in situ*, and with molecular resolution using amplitude modulation (AM) AFM. Typical molecular structures formed by water and alcohol at the surface of HOPG are visible in Fig. 1. In a ternary mixture of water, methanol (MeOH) and ethanol (EtOH) the interface appeared covered by a regular array of longitudinal rows (white arrow in Fig. 1a) running in parallel.

For each row, parallel and perpendicular substructure can be observed: several sub-rows are running in parallel, showing a clear contrast in topography. Finer details can be observed perpendicularly to the rows (dotted white lines in Fig. 1a), with a periodicity of $6.1 \pm 0.2$ Å (Fig. 1b).

To simplify the system and help determination of the molecular arrangement at the interface, we explored in depth the interface formed by HOPG and binary mixtures of water and MeOH. The concentration $X_{MeOH}$ of MeOH in the water influences the type of structure obtained. For most concentrations studied $(0.1 \leq X_{MeOH} \leq 0.7)$, the interface developed a regular array of longitudinal rows similar to that observed in the ternary mixture, and with a spacing of $58 \pm 2$ Å between rows (white arrow Fig. 1c). The direction of the rows is epitaxially determined by the underlying HOPG lattice (Supplementary Figs 1 and 2) and they typically form domains spanning tens of micrometres, with different domains oriented at 120° with respect to each other (Supplementary Fig. 1). As in the case of the ternary mixture, substructure can be observed for each row with several sub-rows running in parallel. Although less clear, finer details can be also perceived perpendicularly to the rows (dotted white lines, inset in Fig. 1c). With a periodicity of $5.9 \pm 0.6$ Å (Fourier analysis, Supplementary Fig. 2), these features are too large for the HOPG itself and can be assigned to the self-assembly of the liquid molecules perpendicular to the direction of the main rows (along the dotted lines in Fig. 1a,c). The alignment is confirmed by contact-mode imaging that reveals the HOPG atomic lattice underneath the structures (Supplementary Fig. 3).

At lower MeOH concentration $(X_{MeOH} \leq 0.1)$, complex 2D assemblies form, together with unstructured regions of the interface (Fig. 1b). Rows can also be seen occasionally, but far less frequently than at higher alcohol concentration, and only covering a small fraction of the interface. The 2D structures exhibit local periodicity but with a unit cell typically spanning several nanometres. Working in pure water did not yield any structure or periodic features (Supplementary Fig. 4), but exposure of the water to MeOH vapour during imaging (not directly in contact with the imaging water) eventually created small structured domains, some of which exhibit rows (Supplementary Fig. 5). Similarly, working in pure MeOH vapour, in a sealed environment previously dried with pure nitrogen revealed no interfacial structures (<5% humidity, Supplementary Fig. 4).

Once formed, the structures were remarkably stable over a broad range of temperatures. Swapping MeOH for EtOH did not create any ordered 2D structure for a 1:1 concentration, despite the verified presence of a stable molecular layer on the surface (Supplementary Fig. 6), consistently with previous studies[11].

The dependence of the different interfacial structures observed on the MeOH concentration, the need for both water and MeOH, and the absence of ordered structures in EtOH at similar concentration all indicate that the structures are specific molecular assemblies that incorporate both water and MeOH molecules. Epitaxial effects from the HOPG are also important as evidenced by the symmetry of the structures.

**Atomistic and molecular dynamics simulations**. To elucidate the precise molecular organization of the structures and gain insights into the system's formation, we conducted atomistic computer simulations. The size and time scales involved in the nucleation process make it impossible to treat explicitly the electronic structure of the problem. Instead, we opted for a multi-scale approach, in which we first parameterized empirical interactions based on *ab initio*, DFT calculations, which were then followed by classical molecular mechanics simulations on a much larger scale.

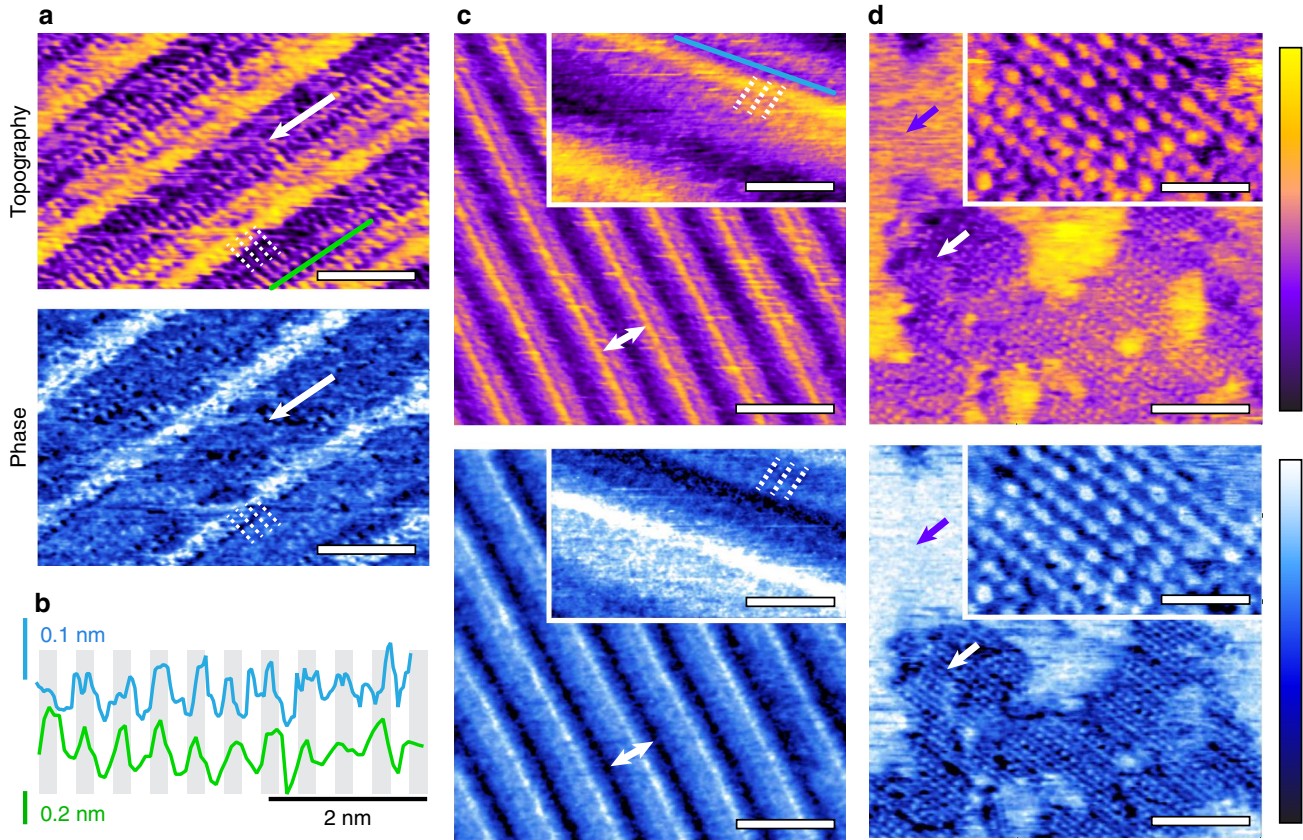

**Figure 1 | High-resolution AM-AFM imaging of the HOPG–liquid interface.** (**a**) Regular array of longitudinal rows (white arrow) obtained in a 1:1 water-MeOH mixture spiked with <1% EtOH. Each row is composed of several ~5 Å wide sub-rows running in parallel, and epitaxially following the underlying HOPG lattice (Supplementary Fig. 3). Finer structure with 6.1 ± 0.2 Å periodicity can also be seen perpendicular to the rows (dotted white lines), as evidenced by the green profile in (**b**) where the periodicity is highlighted. (**c**) A similar row-like structure is visible in a 1:1 water-MeOH solution without any EtOH. The inter-row distance (white arrow) is 58 ± 2 Å. Magnification over a set of row (inset) shows a 5.9 ± 0.6 periodicity perpendicular to the rows (dotted white lines), but less marked than in (**a**). A profile taken along the row (blue in **b**) confirms the finer structure although with an amplitude ~4 times less marked than for (**a**). The clearer resolution with EtOH may be due to the slightly larger periodicity making the difference close to the resolution limit for this system. The structures could be observed with different cantilevers and types of AFM. At low (~5%) MeOH concentration (**d**), other 2D structures can be observed forming islands (white arrow) in an otherwise unstructured interface (purple arrow). These structures require a higher temperature to nucleate. Magnification over these assemblies (inset) shows a network of protrusions organized in alternated rows. Along a row, the distance between the smaller protrusions is ~5.5 Å and between the larger protrusions is ~11 Å. The distance between adjacent rows is ~6.3 Å. The assemblies tend to exhibit a high degree of polymorphism at the molecular level (Supplementary Fig. 2). The scale bars are 20 Å (a), 100 Å (c,d), 30 Å (c,d, inset). The purple colour scale bar represents topographic variations of 8 Å (**a**), 8 Å (**c**), 5 Å (**c**, inset), 12 Å (**d**), and 7 Å (**d**, inset). The blue scale bar represent a phase variation of 15° (**a**), 12° (**c**), 10° (**c**, inset), 13° (**d**), and 16° (**d**, inset). The temperature is 36 ± 3 °C (**a–c**) and 60 ± 0.1 °C (**d**).

We performed simulations using a super-cell geometry comprising eight graphite layers aligned along the $xy$ plane and separated by the MeOH-water mixture. The simulation cell included about 40,000 atoms (Fig. 2a). Simulations were performed for five different $X_{MeOH}$ compositions, and at constant temperature and pressure, leaving only the $z$ direction free to fluctuate.

We consistently observed the formation of a strongly structured surface layer, with preferential segregation of MeOH molecules in the first layer above the surface (Fig. 2b). The results are qualitatively similar for all $X_{MeOH}$ (Supplementary Fig. 7) and we hence only discuss here the case of $X_{MeOH} = 0.5$. The DFT simulation conducted on a smaller box (see Methods) produced a density profile in good agreement with the empirical force field results, validating our simulation strategy. We note that the higher fraction of MeOH molecules observed here for all $X_{MeOH}$ at the interface is comparable to experimental results at the water–vacuum interface[20], suggesting that the surface segregation is a general feature of water-MeOH mixtures at hydrophobic interfaces.

The dynamics of the first surface layer is extremely slow, with a residence time for single MeOH molecules exceeding several tens of nanoseconds (Fig. 2c). This dynamics is hence inaccessible to direct first-principles simulations. A MD simulation run of 100 ns did not capture the formation of any long-lived stable structure at the interface. However, this does not necessarily indicate a deficiency of the model, but rather provides an indication of the long time scale involved in the dynamics of this system and is consistent with the activated nature of the process by which the experimentally observed 2D pattern is formed. We also note that the affinity shown by MeOH for the disordered interface layer at all concentrations might only reflect the initial stage of formation of the structures which may subsequently differ in their final, well-defined stoichiometry.

Although simulations cannot directly capture the formation of the ordered pattern, they can provide important clues about its dynamics and periodicity. First, the potential of mean force for a water and MeOH molecules relative to the underlying HOPG lattice shows corrugations that are very small (Fig. 2d). The presence of weak interactions with the substrate is consistent with

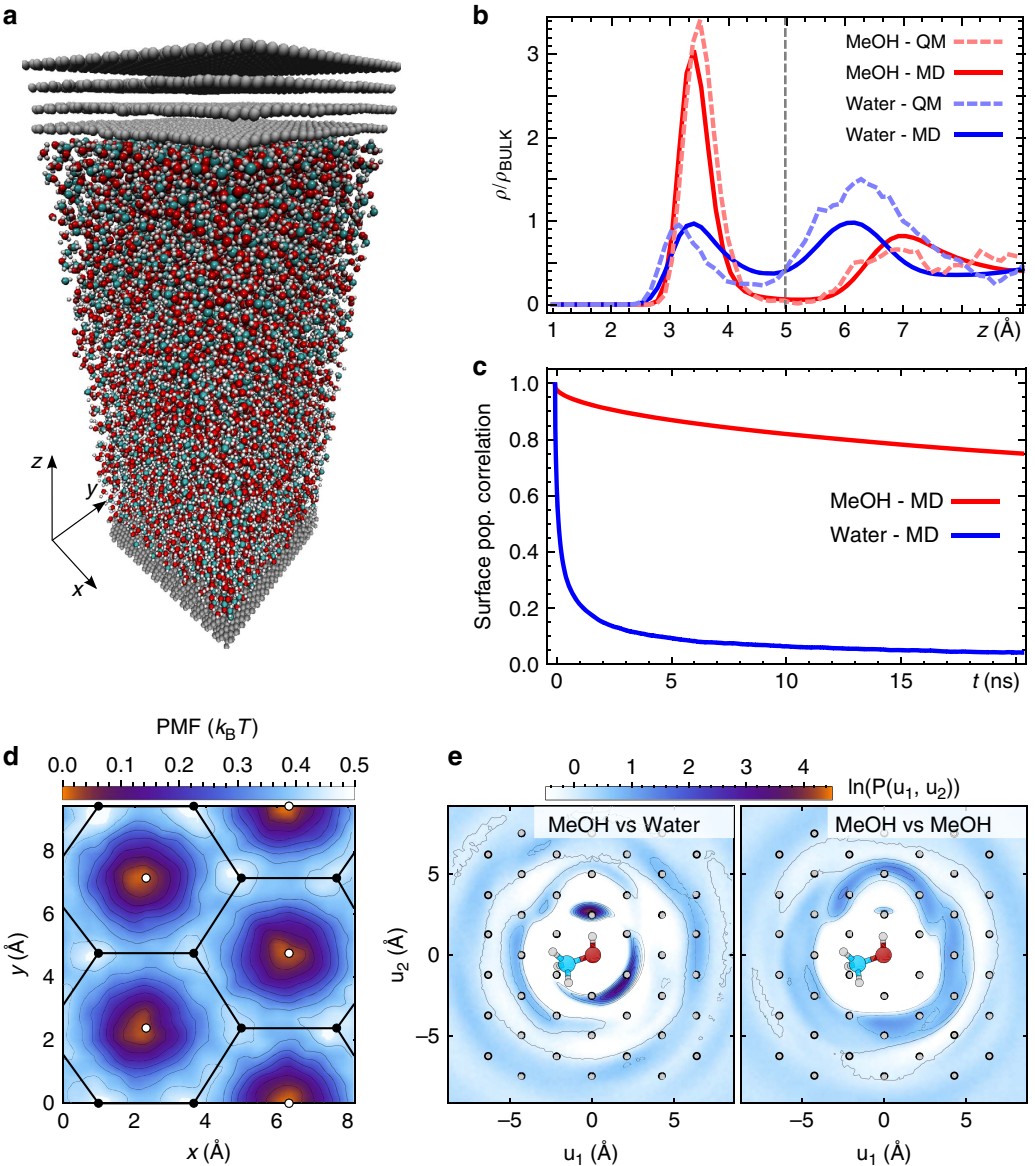

**Figure 2 | MD simulations of the water-MeOH mixture at the interface with HOPG.** The simulations were performed to explicitly model mixtures of different concentrations on the surface of graphite. (**a**) A snapshot of the ($50 \times 50 \times 150$ Å) simulation supercell, using a slab geometry with periodic boundary conditions. (**b**) The density profile along $z$ for a 1:1 water:MeOH mixture shows strongly structured water and MeOH layers between 1 and 5 Å. The distributions obtained with an empirical force field are in good agreement with reference *ab initio* calculations on a smaller supercell (Supplementary Fig. 6). (**c**) The relaxation time for molecules in the first layer above the surface is in the order of several tens of nanoseconds. (**d**) The spatial distribution of MeOH is consistent with a corrugated potential of mean force (PMF) in the $xy$ plane, commensurate to the graphite lattice. This in-plane corrugation is however much weaker than the potential of mean force along $z$ (in **b**). (**e**) In-plane oxygen–oxygen distribution function around a reference MeOH molecule for water (left) and MeOH (right). The reference MeOH molecule is centred with its OH bond aligned along the vertical axis. Water shows an increased propensity for being in the first coordination shell while MeOH sits mainly in the second coordination shell. White dots correspond to the underlying graphite lattice. We note a small mismatch between the characteristic length scale of the water-MeOH H-bond network and the periodicity of the HOPG surface.

the experimental observations, in particular the partial alignment of the rows' substructure with the underlying HOPG lattice (Fig. 1a). Second, a remarkable degree of local conformational ordering is visible. The in-plane oxygen–oxygen distribution function around a given MeOH molecule demonstrates a strong topological correlation (Fig. 2e)—the nearest neighbour of a MeOH molecule tends to be a water molecule and *vice versa*, while the second neighbour tends to be a molecule of the same species. Wires composed of hydrogen-bonded alternating MeOH and water molecules are present in the bulk solution[13–16] (Supplementary Fig. 8), but much more pronounced at the HOPG surface.

To identify stable 2D monolayer structures at the interface, we used replica exchange simulations coupled with geometry optimization at regular intervals (see Methods). By considering even the smaller simulation cells—four MeOH and four water molecules—hundreds of thousands of local energy minima were generated, a clear sign of the glassiness of potential energy landscape. This large number of inequivalent structures rendered systematic classification difficult and we used a non-linear dimensionality reduction technique[21] to automatically cluster similar 2D structures together. This approach allowed us to identify a specific group of structures exhibiting consistently lower energies (Fig. 3a), all based on a characteristic square motif formed by two water and two MeOH

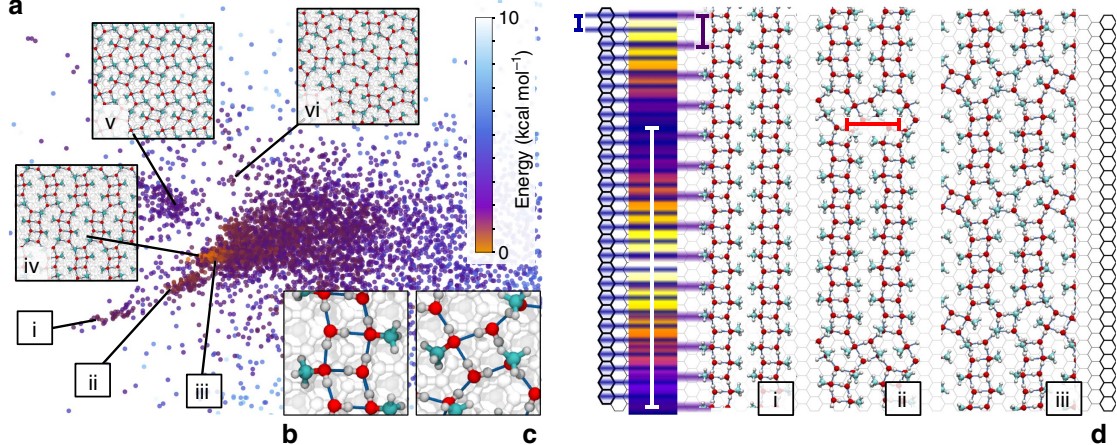

**Figure 3 | Catalogue of stable 2D structures for a 1:1 MeOH:water monolayer.** Each structure was obtained by replica-exchange simulations (**a**). Each dot represents a particular structure whose stability (energy) is colour-coded. Six representative structures (i–vi) are presented (in inset (**a**) and in (**d**)). Most structures share a recurring water-MeOH 'square' motif (**b**) forming ribbons. The same motif exists with a corner-sharing arrangement (**c**), inducing defects in the ribbon structure (**d**) that relieve the mismatch between the periodicity of the ribbon (5.20 Å, purple bar) and that of HOPG underneath (2.46 Å, blue bar). This mismatch induces Moiré patterns with a lengthscale of ~45 Å (white bar). The yellow-purple ribbon in (**d**) illustrates the degree of mismatch using the colour scale of AFM topographic images. The distance between adjacent ribbons is typically 8.2 Å (red bar).

molecules (Fig. 3b,c). This molecular unit can combine in two ways—either sharing a side, thereby forming one-dimensional ribbons (Fig. 3b), or sharing corners (Fig. 3c).

The row structures observed experimentally can be explained as a combination of ribbon segments oriented perpendicular to the main rows (in Fig. 1a) and corner-sharing defects that make it possible to relieve the mismatch between the graphite lattice and the self-assembled layer. The main rows are due to a Moiré pattern that originates from a mismatch between the periodicity of the molecular arrangement along a ribbon (5.20 Å) and that of the underlying graphite substrate (2.46 Å). The resulting Moiré periodicity (~45 Å, Fig. 3d) is comparable to that of the row spacing observed by AFM, and consistent with the substructure running parallel to each row. The substructure appearing perpendicular to each row in the AFM images can be interpreted as single ribbons. Simulations also predict different molecular arrangements with similar energies (Fig. 3a) due to the presence of motifs forming basic 'building blocks'. This is consistent with the experimental observation of multiple regular 2D structures (Fig. 1b) often with a high degree of polymorphism (Supplementary Fig. 2).

While there is a remarkable consistency between simulations and experiments, the comparison remains mostly qualitative. On the one hand, the periodicity of the Moiré pattern is very sensitive to the length scale of the ribbons: reducing by 1% the size of the square repeat unit would suffice to obtain perfect agreement with the experimentally-observed spacing of the rows. On the other hand, the simulation of the monolayer is at constant coverage and neglects interactions with the bulk liquid, which leads to significant over-estimation of the spacing of ribbons in the perpendicular direction.

A search for stable 3D or multi-layered structures by simulations did not yield clear insight due to the large number of possible conformations. Nonetheless, introducing the bulk solution over some of the lowest-energy monolayer structures did not alter their stability for several tens of nanoseconds at 280 K (effectively 30 K above the melting temperature of the water model).

**A thermally nucleated process.** The slow interfacial dynamics observed in simulations hints at a nucleation-based process that may be observable by AFM whose time resolution is typically in the millisecond to second domain. Experimentally, the interfacial structures were seen to form through thermally-activated nucleation and subsequent growth. An example of such growth is presented in Fig. 4a–d at 40 °C. For MeOH concentrations $0.1 \leq X_{MeOH} \leq 0.7$ we found a nucleation temperature $T_n = 35 \pm 5$ °C, determined as the lowest temperature where nucleation could be observed in a matter of minutes. Higher temperatures accelerated the process, while at lower temperatures no nucleation was observed over hours.

During growth ($X_{MeOH} \geq 0.1$), the whole surface becomes first covered with row-like structures that nucleate from small islands (Supplementary Fig. 10). The ordered layer can subsequently develop over multiple levels if the temperature is high enough (Fig. 4e). The higher levels often exhibit gaps and holes with edges parallel and perpendicular to the rows, consistent with the substructure presented in Fig. 1a and with the model proposed from the simulations (Fig. 3d). To create >2 layers, it was necessary to heat the solution above 45 °C. For $X_{MeOH} \leq 0.1$ temperatures higher than 35 °C were needed to start the nucleation process. At high temperatures (70 °C), the second layer could occasionally be seen growing on the 2D lattice domains, but this layer could be easily removed by the tip (not shown). All structures shown here were fully stable once formed, and remained unaltered when the sample was subsequently cooled down.

## Discussion

The experimental results and the simulations provide a consistent picture of the interface between the HOPG and the solution: the presence of a hydrophobic surface creates layering of MeOH molecules near the interface, with a strong topological but short-ranged order. At lower temperatures, the interfacial liquid is glassy and the lateral organization of molecules is mostly amorphous. Increasing the temperature allows the system to overcome the layer's glassiness and a 2D self-assembly of water and MeOH spontaneously nucleates above a system-specific temperature $T_n$. The cohesion of the structures is ensured by an extended 2D hydrogen bond network that alternates water and MeOH molecules. This interplay between a frustrated interfacial H-bond network and the structure of the solid surface gives rise

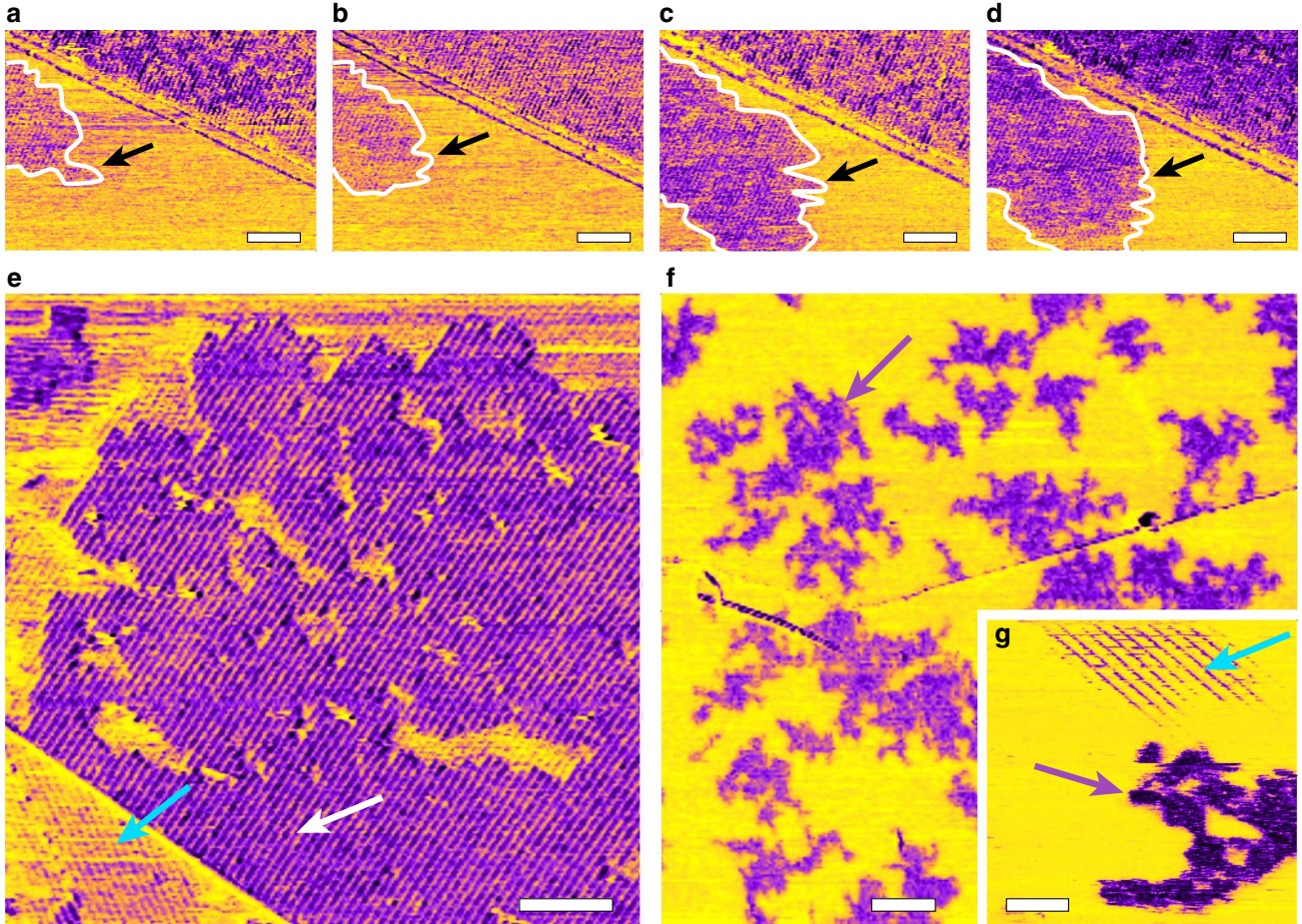

**Figure 4 | Thermally-activated formation of organized water-MeOH structures.** All images are phase images, displayed in the topography colour scale only in this figure for better contrast. The full set of data is available in Supplementary Fig. 9. The growth of row domains can be followed in real time (**a–d**). The temperature is $40 \pm 0.1\,^{\circ}$C (measured in the metal piece supporting the HOPG) and the time between consecutive frames is 36 s. The edge of the growing domain is delineated in white and indicated with a black arrow. Once the surface is fully covered with rows (blue arrow in **e**), the second layer of rows (white arrow) can form on top of the first layer. Its thickness is about 2.9 Å (Supplementary Fig. 11). The process is enhanced by increasing temperature (here $45 \pm 0.1\,^{\circ}$C). At low MeOH concentration (here 5%) (**f**) higher temperatures are required to nucleate stable interfacial structures, which do not cover the whole HOPG surface (purple arrow). Higher magnification (**g**) reveals the coexistence of rows (blue arrow) and 2D structures such as those shown in Fig. 2d (purple arrow). The temperature is $60 \pm 0.1\,^{\circ}$C in (**f**) and (**g**). The scale bars are 50 nm (**a–e**), 100 nm (**f**) and 20 nm (**g**).

to the different patterns observed, depending on the composition of the liquid. Due to the formation of Moiré patterns with HOPG, it is difficult to accurately derive a thickness value for the structured interfacial layer. However, we occasionally observed a second layer developing atop the first layer at higher temperatures, with in one case up to three layers present at 55 °C (Supplementary Fig. 11). The second and third layers were incomplete and with many gaps, providing the opportunity to quantify the layers thickness. We measured a value of ~ 2.9 Å in all cases, consistent with the proposed molecular model (Supplementary Fig. 11b–c).

We would like to emphasize the fact that the stable structures presented in this paper are fundamentally different in nature than those formed by longer alcohol molecules such as dodecanol at hydrophobic interfaces, when in contact with the liquid at room temperature[19]. If the alkyl chain of the alcohol molecules is sufficiently long, van der Waals interaction between adjacent alkyl chains and with the hydrophobic graphite is sufficient to spontaneously form self-assembled monolayers at the graphite–liquid interface[19]. The shorter the chain, the lower the temperature required for a monolayer to form, with pentanol layers melting already around 200 K ( − 73 °C) on graphite[19]. Van

der Waals-stabilized MeOH or EtOH layers would therefore not be possible inside the liquid at room temperature; water, EtOH and MeOH can form layers on graphite but below 200 K (ref. 9). Long-lived stable interfacial structures were only observed under confinement[5,6,22], or at low temperature[12,23]. Our results show that by incorporating both water and alcohol in the interfacial structures, it is possible to create an H-bond dominated assembly that is considerably more robust than those relying on van der Waals interactions, and that thrives above 300 K. This is illustrated by the successful use of the interfacial structures as template for directed ionic self-assembly under an external electrical potential (Supplementary Figs 12 and 13).

We note that in vacuum, alcohol monolayers on graphite tend to melt at higher temperatures than in liquid, for example around 260 K for pentanol[24]. A 'confinement' of interfacial MeOH between the graphite and the bulk water could, in principle, help explain the observed structures and increased melting temperature. However, since both water and MeOH are present in the structures, this interpretation would be limited to the thermodynamics driving the self-assembly, not the resulting structures.

AFM results show that the solid-like interfacial layer is formed by thermal nucleation. In the presence of an ordered substrate

such as HOPG, the regular arrangement of the water and alcohol molecules at the interface can form a Moiré pattern, a phenomenon well-known for self-assembled monolayers on HOPG (ref. 25). Depending on the composition of the bulk solution, different 2D molecular arrangement can be observed at the interface. The most common assembly ($0.1 \leq X_{MeOH} \leq 0.7$) nucleates around $\sim 35\,°C$, and subsequently grows to cover the whole interface (Fig. 1c, Supplementary Fig. 1). The simulations show that the observed 2D arrangements are predominately formed by water-MeOH molecular wires lying flat and parallel on the surface and stabilized through collective effects. Working at lower MeOH concentration forms different patterns that do not cover the whole interface and exhibit a high degree of heterogeneity at the nanoscale (Supplementary Fig. 2e–f). Higher nucleation temperatures are also required, suggesting that low alcohol concentration favours assemblies corresponding to local energy minima (Fig. 3a) rather than the more stable row-like structure. The high degree of polymorphism in the molecular arrangement could also explain the difficulty for nucleated domains to grow over large areas.

Simulations show that the interaction potential between the liquid and the HOPG surface is relatively weak, and the existence of recurrent molecular motifs suggests that this type of molecular assembly is not specific to HOPG or to MeOH, but can in principle occur at any hydrophobic interface and with different alcohols.

We note that the row-like structures reported here bear some resemblance with recent reports of longitudinal nanobubbles forming at the surface of HOPG in pure water, allegedly from dissolved gas molecules[26,27]. Nanobubbles can be excluded here considering the scale of the smallest features observed. Any bubble that size would experience considerable Laplace pressure and rapidly coalesce into larger bubbles[26,28,29]. Additionally, thorough degasing of the water-alcohol mixture had no noticeable effect on the structure and their formation.

In conclusion, we show that solutions containing mixtures of water and simple alcohols can form stable, solid-like 2D molecular assemblies at hydrophobic interfaces. The low mixing entropy of the solution[13] favours molecular arrangement that alternate alcohol and water molecules, allowing the formation of transient supramolecular assemblies in the bulk liquid. At the interface with hydrophobic solids, these transient molecular assemblies can nucleate stable structures whose integrity are maintained by an extended network of hydrogen bonds. The structures reported here are different from the well-known molecular assemblies formed by surfactant or alkanes on HOPG where the force driving the self-assembly is the van der Waals interaction between long carbon chains and the HOPG's surface. In the case of surfactants, the molecules are arranged into hemimicelles that follow the HOPG lattice[30], while alkanes tend to lie flat on the surface[31] or as monolayers[19], with predictable molecular arrangement. Here neither water nor alcohol single molecules remain permanently adsorbed at the interface between the liquid and graphite at room temperature, but a collective effect is necessary. This is in contrast to previous reports of hydrogen-bonded molecular arrangement at interfaces that involved relatively large molecules able to adsorb permanently on the surface[32].

We expect that the existence of stable water-alcohol assemblies at hydrophobic interfaces will have important consequences for a wide range of interfacial processes where mixtures involving water and alcohol play a central role, for example in electrochemical processes[33], graphene-based technology including alcohol fuel cells and catalysis[34], food processing[35], health science[36], chemical synthesis[37], controlled molecular self-assembly[38] and surface treatment[39].

## Methods

**Sample preparation.** The liquid mixtures were prepared using HPLC-grade alcohols (purity $> 99.9\%$) (Sigma-Aldrich, Dorset, UK) and ultrapure water ($18.2\,M\Omega$, Merck-Millipore, Watford, UK). In a typical experiment, water and alcohol were mixed with the desired proportions in a 20 ml glass vial. The vial was first thoroughly cleaned with the alcohol of interest and subsequently with ultrapure water and dried under a flow of nitrogen. The mixture was then degassed by sonication for 5 min. The degasing step was conducted to limit the formation of bubbles on the graphite surface[26,27,29], but omitting the step did not lead to significantly different results. The substrate (HOPG, SPI supplies, West Chester, PA) was glued with epoxy resin (Araldite, RS components, UK) onto a stainless steel disc (SPI supplies) and allowed to cure for 24 h. The HOPG was pressed against the metal disc to obtain a satisfactory thermal contact. The HOPG was then mounted on a hot plate and allowed to heat ($> 100\,°C$) for several minutes to evaporate possible traces of solvent on the graphite. Before each experiment, the HOPG surface was cleaved several times. A drop (100 μl) of liquid was deposited on the freshly cleaved surface, and the sample immediately placed into the AFM's imaging cell. The imaging cell was then promptly sealed to limit evaporation (Cypher). Such a degree of control was not possible with the Multimode AFM.

To ensure reliability of the results, experiments were reproduced in different laboratories, using solvents from different production batches and water purification systems.

**Atomic force microscopy.** All atomic force microscopy results (except Fig. 1a and Supplementary Figs 11 and 12) were obtained on a commercial Cypher ES system (Asylum Research/Oxford Instruments, Santa Barbara, CA) equipped with photothermal excitation and operated with the tip/sample fully immersed in liquid. The cantilevers (Arrow UHF-AUD, Nanoworld, Neuchatel, Switzerland and Olympus RC800 PSA, Olympus, Tokyo, Japan) were calibrated using their thermal spectra ($k \sim 2$–$3\,N\,m^{-1}$ and $k \sim 0.7\,N\,m^{-1}$, respectively). The liquid cell was thoroughly cleaned in the imaging solution before each experiment and the graphite substrate freshly cleaved. To achieve reproducible results, it was necessary to avoid any diffraction of the laser used to detect the motion of the AFM cantilever over the sample (Supplementary Fig. 14). This laser being in the infrared, it can locally heat the graphite and induce nucleation. No nucleation could be observed over several hours of scanning, while the sample/liquid temperature was kept below $35\,°C$ (5–$35\,°C$). As soon as $T_n$ was reached, small domains appeared within minutes and started to grow with a rate depending on temperature. The AFM was operated in amplitude modulation with working amplitudes $A$ between 0.5 and 1 nm and a setpoint ratio $A/A_0 > 0.7$, where $A_0$ is the free vibration amplitude of the tip away from the interface. In these conditions the phase lag $\varphi$ between the driving vibration and that of the tip is sensitive to the behaviour of the liquid expelled by the vibrating tip[40,41]. The imaging is dominated by the molecular arrangement of the liquid at the interface with the solid with the imaging phase particularly sensitive to variations in the molecular orientation and exposed chemical groups of the adsorbed liquid[42,43]. The resolution is enhanced by short-range solvation forces[44–47] and high-resolution images could routinely be achieved. The nucleation of interfacial structures was controlled using an in-built heating/cooling system with an experimental accuracy better than $\pm 0.1\,°C$. The temperature measured is that of the metal part supporting the HOPG substrate. Not only temperature, but also heating/cooling rates are measured so as to ensure thermal equilibrium is reached. Generally, the sealed environment of the imaging cell allowed for stable imaging at higher temperature ($> 40\,°C$) for hours. Using this setup, the nucleation process described in Fig. 4 was fully reproducible. Experiments in pure MeOH are challenging given the solvent's high vapour pressure, which induced poor thermal stability. We therefore conducted experiments in saturated MeOH vapour using a sealed cell previously dehydrated with pure nitrogen (Supplementary Fig. 4).

A small part of the results presented (Fig. 1a and Supplementary Figs 12 and 13) were obtained with a Multimode IIIA system (Digital Instruments, now Brucker, Santa Barbara, CA), and a different batch of chemicals to ensure reliability of the results. The same imaging conditions were used as with the Cypher, but without thermal control. The interfacial structures formed after various time intervals, probably depending on the alignment of the laser on the cantilever. An external DC power generator (TTi instruments, Hutingdon, UK) was used for the electrical measurement presented in Supplementary Figs 12 and 13. A contact was made with the HOPG using silver paint and its quality was checked with an ohmmeter (resistance $< 5\,\Omega$). The counter electrode consisted of a copper wire looping around the cantilever. Since the purpose of this measurement was only to serve as proof of principle, no reference electrode was used.

**Ab initio simulations.** To (partially) bridge the gap between experimental and computationally accessible time scales, we have adopted a multi-scale strategy in which we combined empirical forcefield simulations with validation by *ab initio* electronic structure calculations based on DFT. We performed all *ab initio* simulations using the Quickstep module of the CP2K electronic structure package[48]. We used the BLYP functional[49,50], combined with a pseudo-potential formalism[51], expanded the valence Kohn–Sham orbitals in a DZV atomic basis set, and used a plane-wave cutoff of 300 eV to represent the density. Dispersion corrections were included with a semi-empirical term based on the D3 framework[52]. Within this

**Table 1 | Best fit parameters describing the interactions between atoms.**

|  | $\varepsilon$ (Kcal mol$^{-1}$) | $\sigma$ (Å) | $\alpha$ |
|---|---|---|---|
| O(Water)-C(Graphite) | 0.01000 | 3.870 | 8.294 |
| H(Water)-C(Graphite) | 0.07222 | 2.281 | 12.00 |
| C(MeOH)-C(Graphite) | 0.2122 | 3.694 | 14.82 |
| O(MeOH)-C(Graphite) | 0.1335 | 3.663 | 9.477 |

setting, we ran *ab initio* molecular dynamics (AIMD) trajectories for water, MeOH and a 1:1 water-MeOH solution on top of a graphite slab, using i-PI as the dynamics driver[53], using a time step of 1.0 fs and enforcing constant-temperature sampling at 300 K in the NVT ensemble using a stochastic velocity rescaling thermostat[54] with a relaxation time of 20 fs. Each simulation was ran for about 100 ps, including 10 ps that were discarded for equilibration, and included four graphite layers, with 32 atoms each, and 30 molecules.

**Empirical force-field simulations.** We described interactions between water and MeOH molecules using standard potentials[55–57], and used a Tersoff-kind potential to describe interactions within graphite planes, supplemented with a Lennard–Jones potential to model dispersion interactions between planes[58,59]. The interaction between molecules and the graphite were obtained by fitting a modified Lennard–Jones formula, where we used the standard $r^{-6}$ attractive term, and left the exponent for short-range repulsion as a fit parameter $\alpha$, together with the usual energy and range parameters $\varepsilon$ and $r$, as shown in equation 1:

$$V[r] = 4\varepsilon \left[ \left(\frac{\sigma}{r}\right)^{\alpha} - \left(\frac{\sigma}{r}\right)^{6} \right] \qquad (1)$$

In order to describe both the position and the orientation of the molecules relative to the surface, we included four separate terms: between the O and H of water and the C atoms of graphite, and between the O and C atoms of MeOH and the C atoms of graphite.

Parameters were estimated by fitting the adsorption energies, $\Delta E_{ads}$, of 1,200 different geometries of a single water and a MeOH molecule adsorbed on graphite, computed with the same ab initio set-up described above. The relative RMS error is given by equation 2:

$$\chi^2 = \sum_n \left( 1 - \frac{\Delta E_{ads,n}^{abi}}{\Delta E_{ads,n}^{eff}} \right)^2 \qquad (2)$$

$\chi^2$ is summed over all $n$ configurations and was minimized down to $<10^{-4}$ using the Nelder–Mead Simplex algorithm, which we implemented in i-PI (refs 53,60). The best-fit parameters are reported in Table 1.

Using this semi-empirical framework, we have used the MD package LAMMPS (ref. 61) to perform about 100 ns of rigid-molecules[62] MD trajectories with a time step of 2.0 fs in the *NpT*-ensemble at 300 K and 1 bar, using a Nose–Hoover thermostat[63] with a relaxation time of 200 fs for the temperature and 2 ps for the pressure.

**Monolayer structure prediction and sketch-map.** Monolayer minimum-energy configurations were generated by performing 5 ns of parallel tempering[64] (PT) MD, using replicas distributed between 150 K and 450 K and different sizes supercell (for example $4 \times 4$, $4 \times 16$, $8 \times 8$, $16 \times 4$ and $16 \times 16$ orthorhombic cell of graphite, with four layers along the $z$ direction), using a full coverage of the surface at 1:1 concentration (4 + 4 molecules for the $4 \times 4$ box, and larger numbers of molecules for larger boxes). We stored a snapshot from all replicas every 50 ps of simulation, and performed a conjugate gradient optimization of the monolayer energy. These optimized structures were then processed to generate the sketch-map representation reported in Fig. 3.

Distances between different structures were computed using a REMatch-SOAP kernel[65], with a cutoff of 5 Å, using as environment centres only the O atoms, and using a regularization parameter of 0.01. On the basis of kernel distance we first selected 200 landmarks using a farthest point sampling strategy, and optimized a sketch-map[21,66,67] with the parameters $\sigma = 0.4$, $A = 12$, $a = 2$, $B = b = 12$. We then projected the remaining configurations using an out-of-sample embedding procedure.

Proximity on the map corresponds to structural similarity of the monolayer structures. Such an intuitive representation allowed us to identify the building blocks and assembly rules for the hydrogen-bond network of a monolayer, and the relation with the underlying HOPG lattice.

**Data availability.** The data that support the findings of this study are available from the corresponding author on request.

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

## Acknowledgements

K.V. acknowledges funding from the European Commission (MC CIG 631186) and the Swiss National Science Foundation (Subsidy 200021_140533). M.C. acknowledges generous allocation of computer time from CSCS under the project s466, and funding by the European Research Council under the European Union's Horizon 2020 research and innovation programme (grant agreement no. 677013—HBMAP). FS acknowledges funding from the Swiss National Science Foundation Division II.

## Author contributions

K.V. designed the experiment and discussed them with F.S. K.V. conducted most of the experimental measurements and wrote the paper with input from all authors. J.J.S. contributed to the experiments. D.G. and M.C. conducted the computer simulations. All authors commented on the manuscript.

## Additional information

**Competing financial interests:** The authors declare no competing financial interests.

**How to cite this article**: Voitchovsky, K. *et al.* Thermally-nucleated self-assembly of water and alcohol into stable structures at hydrophobic interfaces. *Nat. Commun.* **7**, 13064 doi: 10.1038/ncomms13064 (2016).

