## [Peer Review File · Nature Communications]

Reviewers' comments:

Reviewer #1 (Remarks to the Author):

The key finding in this paper is that water-methanol mixtures form solid structures at the interface between the liquid phase and a graphite crystal. These structures appear to nucleate and remain stable at relatively high temperatures.

These are interesting and novel observations. One question is to what degree this is actually a new discovery. Pure alcohols form crystalline 2D structures at the vacuum-graphite interface at temperatures far above the bulk melting point of the alcohol (e. g. Morishige and Sakamoto J. Phys. Chem. 103 (1995) 2354). This is also observed at the liquid alcohol-graphite interface where crystalline layers are formed (e.g. Messé et al. J. Colloid. Interface Sci. 266 (2003) 19). Other studies observed interesting Moiré patterns for self-assembled alcohol layers on graphite using STM (e.g. Silly, Nanotechnol. 23 (2012) 225603). Water also forms stable solid structures on the surface of graphite (see work by Bruce Kay and coworkers). In light of these earlier studies it is perhaps not surprising to see the formation of stable alcohol-water structures at the interface. However, the methodology appears to be sound and makes it possible to perform detailed studies of the system at high temperature. The study is carefully carried out and the paper is well-written.

The molecular dynamics simulations help the interpretation of molecular-level details of the experimental observations, but, as also pointed out by the authors, the difference in time scales for MD simulations and experiments makes it difficult to investigate nucleation of structures at the interface. The group may consider employing Monte Carlo methods in future studies to investigate if the used interaction potentials are capable of producing the structures observed experimentally.

To summarize, this appears to be carefully carried out study showing interesting observations of relevance for fundamental understanding and several applied fields.

Reviewer #2 (Remarks to the Author):

Manuscript Number: NCOMMS-16-04615-T

In this study the authors report on surprisingly stable structures of alcohol-water mixtures at hydrophobic interfaces. I find it particularly interesting since they are driven thermally and persist at high temperatures. The AFM studies are complemented with simulations of the system which allow the authors to propose a plausible and consistent structural model. I will recommend the paper for publication after the authors have addressed the following minor comments:

- 1) It is not completely clear from the presentation how many (stable) layers are actually formed and how thick the liquid is on the HOPG. The authors speak of 2D structures but also of different layers. It would be good if this can be summarized at the end.
- 2) The statement that "the liquid is far from homogeneous at the molecular level." at the end of the first paragraph of the introduction requires a reference.
- 3) In the caption of figure 1 please give the experimental conditions, e.g. the water to methanol ratio and the temperature.
- 4) Figure 4, caption. I don't see a purple arrow.

5) Page 11, middle. Figure 5e is mentioned. I assume it should be 4e.

6) Page 3. It is stated that "This prediction could be verified experimentally with AFM..." The "could be" is a little ambiguous and could mean either that it actually *was* verified experimentally or that it *potentially* could be verified experimentally. If it actually was verified by the authors please write that.

7) There are some language issues:

Page 2: "is too small to allow"

Page 3: "differs from previous reports"

Page 3: "can adsorbed"

Page 12: "2D structure is is not"

Page 12: "pressure rapidly coalescence"

Page 12: "the low the"

Page 12: "are arranged in hemimicelles"

Page 13: Ab initio is written as ab-initio interchangeably.

Page 14: "Each simulation was ran for"

Reviewer #3 (Remarks to the Author):

The key result is the observation of ordered structures at a graphite interface in a water-methanol mixture. The data is qualitatively linked with simulations to show a plausible molecular model for the observations.

this is a new result, so is certainly novel, and is of interest because of the importance of the water-alcohol-solid interface in technology and scientific processes.

I do not recommend publication unless some of the AFM data presented is improved as detailed in the below report.

Thermally nucleated self-assembly of water and alcohol into stable structures at hydrophobic interfaces.

Voitchovsky et al

General Comments

In general the paper is interesting and well presented, but I have difficulty convincing myself of two sets of the data. These must be addressed before publication can be considered. Specifically,

- 1) Page 4 with reference to the fine lines with 5.9Å periodicity. You say “although less clear...” well, this is true! The supposed features are hardly observable in the pictures shown (they are obscured even by the white dotted lines drawn to aid the eye) and the Fourier transform also is not particularly convincing either. You must provide a more convincing phase image (zoomed in more, better clarity, something like this). This is important because these apparent ribbon patterns are key to your linking of the computer simulations of Fig. 3 (page 8, last paragraph); without clearer evidence this becomes a bit hollow.
 - As an aside to this can you also comment (in the text on page 4 where you discuss Fig 1) other experimental points related to the small 5.9Å features, namely :
Were these features seen at all $0.1 < X_{\text{meoh}} < 0.7$? were they seen using different tips
i.e. you are sure it is not a tip artifact?
- 2) I have a similar issue with Figure S13c and the related discussion (page 12, 1st paragraph) purporting to show related ordering on PTFE. I cannot see these small features at all and thus cannot agree with your assertion that “These results strongly support the generality of the findings”. You must either show stronger evidence or delete this paragraph and related assertions.

Page 9, lines 5-6. Perhaps I am missing something but I do not see how you link Fig 1b to all the different arrangements seen in Fig 3a. Why would ribbons – the lowest energy structure in your simulations – not also be dominant in Fig 1b as well? Or to put it another way, how does your simulation fold in the fact that Fig. 1b is for low X_{meoh} data?

Minor points

- 1) Page 2, refs 5-6 with water within graphene is probably arising from ions/crystallisation within the water – see follow up papers on this topic.
- 2) Page 2, lines 7-11. Each of these 3 sentences are rather sweeping statements – you need to back up each with some references.
- 3) Page 3, line 7. You say “neither pure alcohol ...” but I assume you do not mean so general and refer just to small chain alcohols like methanol and ETOH. Can you clarify in the text.
- 4) Figure 1. Can you state the temperatures of this specific example. Can you state the concentration for Fig (a).
- 5) Figure 4. Can you state what X_{meoh} is. Also, on line 8 you say “such as those in fig 4”. Is this a mistake as I cannot seem to see what relevance to Fig. 4.
- 6) Similarly, page 11 line 12, where is “Fig. 5e” ??
- 7) Page 12, line 21, do you mean “The low mixing” not “The low the mixing”? or similar? Please clarify.
- 8) Just a note – no comment needed. I think the temperature rise from the laser impinging the HOPG will be rather small so would be surprised if this induces nucleation. Do you have clear evidence for this?

In Supplementary information :

- 1) Page 7, Fig S3. Very hard to see any HOPG lattice in (a). Can you use a higher resolution picture or an insert to show.
- 2) Page 15, line 2. Do you mean “used in Fig. S11”, not “Fig. 5”? A bit confusing.
- 3) Page 17, Fig S14. State what the Temperature is.

Reviewer #4 (Remarks to the Author):

The authors investigated the formation of 2-dimensional structures on graphite surface when a solution of water and alcohol is deposited on it. Using AFM they observe the formation of row-structure domains on the surface. The authors used multiscale molecular dynamics simulations to interpret the AFM image.

The resolution of the AFM images is quite poor. A row-structure can be observed but it is not possible to identify the 2 molecular species in the images. Intermolecular bonds can not be experimentally determined. The calculations are then very speculative: many stable arrangements are possible. Overall the interpretation of the authors is very speculative because the comparison between experimental data and simulation is not convincing.

For these reasons I can not support this article for publication. It should be submitted to a more specialised journal like J. Phys Chem.

In addition the paper is not easy to read because its style is quite confusing. The introduction is for example already discussing the results before they are presented. The structure of the paper should be strongly modified.

REVIEWERS' COMMENTS:

Reviewer #1 (Remarks to the Author):

The authors have improved the manuscript, but a few minor questions remain and changes made to the manuscript raise a few new questions. I suggest that the following comments are considered before publication.

1. Lines 21-22: "Here we show, that mixtures of water with methanol or ethanol can form a novel type of interfaces with hydrophobic solids."

The abstract gives the impression that binary water-ethanol mixtures may also nucleate new structures at high temperatures. This does not appear to be covered in the manuscript and the sentence needs to be rephrased.

2. Lines 63-67: "The structures remain stable at room temperature, pointing to the collective effect of an extended H-bond network. This is significant because most layering of alcohol molecules at interfaces is based on van der Waals interactions^{18,19} which are not strong enough to stabilise short alcohols such as methanol or ethanol at the interface with immersed solids at room temperature."

What do you mean by "stabilize" in the last sentence? The strength of the molecule-surface interactions is similar for the different systems treated here; the question is rather if a specific system will form solid- or liquid-like structures at the interface. Also, polar OH groups play an important role for the intermolecular interactions between small alcohol molecules, making the first part of the sentence unclear in relation to the H-bond network mentioned in the preceding sentence.

3. Lines 73-77 would fit better in a concluding paragraph, or may be removed.

4. Line 109: "...1:1 water-methanol mixture spiked with a small amount of ethanol."

Comments are required about how much ethanol that was actually added to the ternary solution, and why this plays a role for the observed resolution.

5. Lines 176-178: "However, this does not necessarily signal a deficiency of the model, but rather provides an indication of the long time scale involved in the dynamics of this system, and of the activated nature of the process by which the experimentally observed 2D-pattern is formed."

Slow dynamics do not necessarily say anything about the activated nature of the process and the sentence needs to be rephrased.

6. Lines 317-331: It is worth pointing out that an alcohol monolayer at the vacuum-graphite interface often has a substantially higher melting point than the corresponding bulk melting point: ethanol 214 K (bulk 156 K), propanol 230 K (147 K), butanol 255 K (184 K); see Moreshige and coworkers in publications 1990-1999. Methanol is actually an exception with a monolayer melting point of 143 K (bulk 175 K). In addition, it may be worth mentioning that in the water-graphite system investigated here, methanol tends to concentrate at the interface due strong water-water interactions in the bulk and a stronger methanol-graphite interaction compared to water-graphite. The interfacial methanol may therefore be considered as confined between the liquid and solid phase, which obviously results in new solid-like structures and a further increase of the melting temperature.

7. Lines 369-371: "Here neither water nor the alcohol molecules experience an interaction sufficiently strong with the graphite surface to remain stably at the interface at room temperature, but a collective effect is necessary."

The strength of the molecule-surface interactions is similar for the different systems treated here; see comment No 2 above.

8. Some linguistic issues previously listed by Reviewer No. 2 are not yet corrected.

Reviewer #2 (Remarks to the Author):

The authors have addressed my previous comments in a satisfactory way. I now recommend the paper for publication.

Reviewer #3 (Remarks to the Author):

The re-submission is now acceptable for publication.

there is a small correction in the caption of Figure 1.

* At low MeOH concentration (b) other..... I think you mean (d) here.

Reviewer #4 (Remarks to the Author):

The manuscript has been modified.

Some interpretations however still remain quite speculative. For example the direct comparison between AFM images and the proposed modeled structure still remains not so convincing.

In addition in the reviewer 1 comment, the authors wrote that the fundamental difference in their paper with previous results is that the formation of stable 2D structures is not driven by van der Waals interactions between the alcohol alkyl chain and the graphite but rather by an extensive hydrogen bond network between the polar parts of the molecules. This is not really surprising, most of the people would expect this.

Overall I do not think this paper reach the very high standards of Nature Communications. I am still not convinced that this paper represent important advances of significance to specialists within the field.

The original remarks from the reviewers appear in black, our answers in blue and text cited from the revised manuscript is in red.

Reviewer #1 (Remarks to the Author):

The key finding in this paper is that water-methanol mixtures form solid structures at the interface between the liquid phase and a graphite crystal. These structures appear to nucleate and remain stable at relatively high temperatures.

These are interesting and novel observations. One question is to what degree this is actually a new discovery. Pure alcohols form crystalline 2D structures at the vacuum-graphite interface at temperatures far above the bulk melting point of the alcohol (e.g. Morishige and Sakamoto J. Phys. Chem. 103 (1995) 2354). This is also observed at the liquid alcohol-graphite interface where crystalline layers are formed (e.g. Messé et al. J. Colloid. Interface Sci. 266 (2003) 19).

We thank the reviewer for his/her comments and for describing our work as interesting and novel.

The reviewer raises an important point regarding the degree of novelty of our findings in light of earlier studies reporting self-assembly of alcohol molecules at the surface for graphite. The reviewer points out two earlier papers showing self-assembly of alcohol at the surface of graphite in vacuum (Morishige) and interfacial layering of alcohol molecules at the surface of graphite immersed in liquid alcohol (Messé).

We agree with the reviewer that the formation of alcohol layers at the surface of graphite in solution (as described in Supplementary Fig. 6a) is not surprising or novel. Hydrophobic (van der Waals) interactions between the hydrophobic part of the alcohol molecule and the graphite are bound to stabilize and orient alcohol molecules at the interface. This explains, for example, the results of Morishige at 200K and also the fact that alcohols with longer alkyl chains form interfacial layers that are more stable at higher temperatures (Messé).

There is however a fundamental difference with the results presented here: the fact that the formation of stable 2D structures is not driven by van der Waals interactions between the alcohol alkyl chain and the graphite but rather by an extensive hydrogen bond network between the polar parts of the molecules. This has two important consequences: first, both water and alcohol molecules are involved into the 2D assembly. Second, the solid-like layer formed remains stable well above 300K where pure alcohols with longer alkyl chains such as pentanol or heptanol can no longer form stable monolayers (Messé). In fact, neither pure water nor pure methanol can form stable structures in liquid at these temperatures. We believe that this difference with previous work is significant and wholly unexpected firstly because it allows for stable 2D structures to exist in conditions and temperatures where they should normally not exist and secondly because alcohol-water mixtures are central to countless processes.

To better clarify this point, we have added the following paragraph in the discussion, including the two references suggested by the referee:

We would like to emphasise the fact that the stable structures presented in this paper are fundamentally different in nature than those formed by longer alcohol molecules

such as dodecanol at hydrophobic interfaces in liquid at room temperature¹⁹. If the alkyl chain of the alcohol molecules is sufficiently long, its van der Waals interaction with the hydrophobic graphite and with neighbouring alkyl chains is sufficient to spontaneously form self-assembled monolayers¹⁹. The shorter the chain, the lower the temperature required for a monolayer to form, with pentanol layers melting already around 200°K (-73°C) on graphite¹⁹. Van der Waals-stabilised methanol or ethanol layers would therefore not be possible at room temperature. Water, ethanol and methanol have all been shown to form layers on graphite, but in all cases, the layers melt below 200°K⁹. Long-lived stable interfacial structures were only observed under confinement^{5,6,30}, or at low temperature^{12,31}. Here, our results show that by incorporating both water and alcohol in the interfacial structures, it is possible to create an H-bond dominated assembly that is considerably more robust than those relying on van der Waals and thrives above 300°K.

References:

5. Song, J. *et al.* Evidence of Stranski–Krastanov growth at the initial stage of atmospheric water condensation. *Nat. Commun.* **5**, 5837 (2014).
6. Algara-Siller, G. *et al.* Square ice in graphene nanocapillaries. *Nature* **519**, 443–445 (2015).
9. Smith, R. S., Matthiesen, J. & Kay, B. D. Desorption Kinetics of Methanol, Ethanol, and Water from Graphene. *J. Phys. Chem. A* **118**, 8242–8250 (2014).
12. Kong, X., Andersson, P. U., Thomson, E. S. & Pettersson, J. B. C. Ice Formation via Deposition Mode Nucleation on Bare and Alcohol-Covered Graphite Surfaces. *J. Phys. Chem. C* **116**, 8964–8974 (2012).
19. Messé, L., Perdigon, A., Clarke, S. M., Castro, M. A. & Inaba, A. Layer-by-layer surface freezing of linear alcohols at the graphite/liquid interface. *J. Coll. Interf. Sci.* **266**, 19–27 (2003).
30. Bampoulis, P. *et al.* Structure and Dynamics of Confined Alcohol–Water Mixtures. *ACS Nano* (2016). doi:10.1021/acsnano.6b02333
31. Yang, D.-S. & Zewail, A. H. Ordered water structure at hydrophobic graphite interfaces observed by 4D, ultrafast electron crystallography. *Proc. Natl. Acad. Sci. U. S. A.* **106**, 4122–4126 (2009).

Other studies observed interesting Moiré patterns for self-assembled alcohol layers on graphite using STM (e.g. Silly, *Nanotechnol.* **23** (2012) 225603).

We agree with the reviewer that the observation of Moiré patterns on graphite is not new. Several studies have previously reported Moirés involving graphite or graphene for different systems, including alcohol monolayers. Here the existence of a Moiré is not presented as something novel; it is simply an observation. However it provides precious indications about the periodicity of the 2D alcohol-water assembly and helps ascertain the validity of the proposed model. This was perhaps not clear from the text and we re-phrased the corresponding paragraph as follow:

AFM results show that this cohesive layer is formed by thermal nucleation. In the presence of an ordered substrate such as HOPG, the regular arrangement of the water and alcohol molecules at the interface can form a Moiré pattern, a phenomenon well-known for self-assembled monolayers on HOPG³².

32. Silly, F. Moiré pattern induced by the electronic coupling between 1-octanol self-assembled monolayers and graphite surface. *Nanotechnology* **23**, 225603 (2012).

Water also forms stable solid structures on the surface of graphite (see work by Bruce Kay and coworkers). In light of these earlier studies it is perhaps not surprising to see the formation of stable alcohol-water structures at the interface.

Again, we agree with the reviewer about the existence of solid water structures on graphite, such as presented in the work of Kay's lab, or by Yang & Zewail (PNAS 2009) or Kong et al (J. Phys. Chem. C 2012). However, such structures require low temperatures (typically <200K) and the presence of very thin water films with only few water layers. In such cases, the natural confinement of water to the surface of the graphite ensures layering and epitaxial effects often determine the resulting structures. To the best of our knowledge, no stable or ordered water-based structures have been reported for graphite in contact with macroscopic quantities of liquid water at room temperature (300K). The observations reported here are therefore novel, because the interactions dominating the systems previously reported could not explain or predict our findings.

However, the methodology appears to be sound and makes it possible to perform detailed studies of the system at high temperature. The study is carefully carried out and the paper is well-written.

The molecular dynamics simulations help the interpretation of molecular-level details of the experimental observations, but, as also pointed out by the authors, the difference in time scales for MD simulations and experiments makes it difficult to investigate nucleation of structures at the interface. The group may consider employing Monte Carlo methods in future studies to investigate if the used interaction potentials are capable of producing the structures observed experimentally.

We thank the reviewer for this suggestion. We would like to clarify that the reason we focused on molecular dynamics is that simulating the actual nucleation process would require a large-scale model of a liquid, for which molecular dynamics techniques generally provide better sampling efficiency (hybrid Monte Carlo would be a possibility, but would still not make a dent in the several orders of magnitude in time scale that have to be bridged). Simulations of the monolayer were actually computationally efficient thanks to replica exchange, and the challenge was to classify the many local minima, rather than thoroughly sampling the configuration space. We believe that our current analysis is sufficient to give insight on the thermodynamic driving force towards nucleation of an ordered phase, and on the source of the long-range patterns. If we were to attempt a direct simulation of the process in solution, sampling with a bias to encourage ordering would probably be the most promising way to proceed.

To summarize, this appears to be carefully carried out study showing interesting observations of relevance for fundamental understanding and several applied fields.

We thank the reviewer for these positive comments and for his/her useful suggestions. We believe this has helped us clarify important points and improve the

manuscript.

Reviewer #2 (Remarks to the Author):

In this study the authors report on surprisingly stable structures of alcohol-water mixtures at hydrophobic interfaces. I find it particularly interesting since they are driven thermally and persist at high temperatures. The AFM studies are complemented with simulations of the system which allow the authors to propose a plausible and consistent structural model.

We thank the reviewer for these encouraging comments and for sharing our enthusiasm about the reported findings.

I will recommend the paper for publication after the authors have addressed the following minor comments:

1) It is not completely clear from the presentation how many (stable) layers are actually formed and how thick the liquid is on the HOPG. The authors speak of 2D structures but also of different layers. It would be good if this can be summarized at the end.

We agree with the reviewer and have now added a clearer discussion about this issue, supported by a new figure (Supplementary Fig. S11):

Due to the formation of Moiré patterns with HOPG, it is difficult to accurately derive a thickness value for the structured interfacial layer. However, we occasionally observed a second layer developing atop the first layer at higher temperatures, with in one case up to three layers present at 55°C (Fig. S11). The second and third layers were incomplete and with many gaps, providing the opportunity to quantify the layers' thickness. We obtained a value of ~ 2.9 Å in all cases, consistent with the proposed molecular model (Fig. S11b-c).

2) The statement that “the liquid is far from homogeneous at the molecular level.” at the end of the first paragraph of the introduction requires a reference.

The following references have been added to this statement:

1. Dixit, S., Crain, J., Poon, W. C. K., Finney, J. L. & Soper, A. K. Molecular segregation observed in a concentrated alcohol–water solution. *Nature* **416**, 829–832 (2002).
2. Nagasaka, M., Mochizuki, K., Leloup, V. & Kosugi, N. Local Structures of Methanol–Water Binary Solutions Studied by Soft X-ray Absorption Spectroscopy. *J. Phys. Chem. B* **118**, 4388–4396 (2014).
3. Corsaro, C. *et al.* Clustering Dynamics in Water/Methanol Mixtures: A Nuclear Magnetic Resonance Study at 205 K. *J. Phys. Chem. B* **112**, 10449–10454 (2008).

3) In the caption of figure 1 please give the experimental conditions, e.g. the water to methanol ratio and the temperature.

Figure 1 has been modified to better evidence the sub-structure along the main rows. The caption has also been updated and this information added.

4) Figure 4, caption. I don't see a purple arrow.

The colour of the purple arrow was darker than intended; this has now been corrected.

5) Page 11, middle. Figure 5e is mentioned. I assume it should be 4e.

This was indeed a mistake and has now been corrected.

6) Page 3. It is stated that "This prediction could be verified experimentally with AFM..."

The "could be " is a little ambiguous and could mean either that it actually was verified experimentally or that it potentially could be verified experimentally. If it actually was verified by the authors please write that.

The "could be verified" was meant as "we were able to verify". However, Reviewer 3 pointed out that the data supporting this statement (previously Supplementary Fig. S13) was not as clear as for HOPG and suggested we remove it altogether from the paper. We have decided to follow this advice since the results were not fundamental to support our findings. Consistently, the above-mentioned sentence was also removed from the manuscript.

7) There are some language issues:

Page 2: "is to small to allow"

Page 3: "differs from previously reports"

Page 3: "can adsorbed"

Page 12: "2D structure is is not"

Page 12: "pressure rapidly coalescence"

Page 12: "the low the"

Page 12: " are arranged is hemimicelles"

Page 13: Ab initio is written as ab-initio interchangeably.

Page 14: "Ech simulation was ran for"

These mistakes have now been corrected.

We thank the reviewer for this careful reading of our manuscript and suggesting useful changes to improve readability and clarity.

Reviewer #3 (Remarks to the Author):

The key result is the observation of ordered structures at a graphite interface in a water-methanol mixture. The data is qualitatively linked with simulations to show a plausible molecular model for the observations.

This is a new result, so is certainly novel, and is of interest because of the importance of the water-alcohol-solid interface in technology and scientific processes.

We thank the reviewer for recognising the novelty and importance of our findings.

I do not recommend publication unless some of the AFM data presented is improved as detailed in the attached report.

[The report, initially provided in a separate document, has been copied below for convenience]

We have conducted new experiments, with different cantilevers/tips and different atomic force microscopes to ensure reproducibility of our findings and better support our conclusions (mentioned in the caption of the updated Fig. 1). Several figures have been improved or added as a result and the changes are detailed in our point-by-point response to the reviewer's comments hereafter.

General Comments:

In general the paper is interesting and well presented, but I have difficulty convincing myself of two sets of the data. These must be addressed before publication can be considered. Specifically,

1) Page 4 with reference to the fine lines with 5.9Å periodicity. You say "although less clear..." well, this is true! The supposed features are hardly observable in the pictures shown (they are obscured even by the white dotted lines drawn to aid the eye) and the Fourier transform also is not particularly convincing either. You must provide a more convincing phase image (zoomed in more, better clarity, something like this). This is important because these apparent ribbon patterns are key to your linking of the computer simulations of Fig. 3 (page 8, last paragraph); without clearer evidence this becomes a bit hollow.

The reviewer makes an important point here and we agree that it is essential to better present the fine structure perpendicular to the main interfacial rows. To improve this aspect of the manuscript we have taken two separate actions:

1. We have extensively modified Fig. 1. We introduced for comparison an image of the interfacial structure obtained in a methanol-water solution spiked with a small amount of ethanol. We noticed that this considerably enhances the resolution of the smaller features, presumably by slightly altering the interfacial hydrogen bond network. The ~ 0.6 nm feature periodicity is immediately visible in the new Fig. 1a, and now serves as a clear reference. A height profile confirms the presence of the features and the stated periodicity (Fig. 1b). Although the features are clear in the presence of ethanol, the paper's analysis and simulations rely on binary methanol-water mixtures with no ethanol. In such cases, the same features are less clear (Fig 1c), but a height profile taken along the main rows still evidences the ~ 0.6 nm periodicity (Fig. 1b). Profiles taken with and without ethanol are very similar (Fig. 1b) although the amplitude of the periodic oscillation appears about 4 times less important without ethanol, explaining why the features are more faint. We feel that the use of height profile is the best manner to confirm these features without resorting to advanced image processing or filtering which might be more difficult to justify objectively.
2. We conducted more experiments with different types of tips to confirm the presence of these features and rule out any imaging artefact. We were able to image the small features, although often with doubled periodicity (Supplementary Fig. S2c). This is not surprising since the features are close to the resolution limit of the AFM. A comprehensive Fourier analysis averaging over whole regions of the reciprocal space shows clear peaks at ~ 0.6 nm and ~ 1.2 nm (Supplementary Fig. S2d). Generally, the influence of the HOPG substrate and the 'soft' nature of the self-assembled layer (it can be removed by the AFM tip in harsh imaging conditions) makes high-resolution imaging

challenging and the periodicity measured tend to induce broad peaks in the reciprocal space.

The text accompanying both Fig 1 and Fig S2 has been modified to reflect the changes.

As an aside to this can you also comment (in the text on page 4 where you discuss Fig 1) other experimental points related to the small 5.9Å features, namely: Were these features seen at all $0.1 < X_{\text{MeOH}} < 0.7$? Were they seen using different tips i.e. you are sure it is not a tip artifact?

These features are difficult to resolve since they push the AFM resolution to the limit. This is clear in the new Fig. 1 where the features are about 4 times less pronounced in MeOH than in the solution spiked with EtOH. We were however able to observe these features multiple times, on different samples, with different tips and with different AFMs. We have now added some more images in Supplementary Information (Fig. S2) and clarified the text in the caption.

2) I have a similar issue with Figure S13c and the related discussion (page 12, 1st paragraph) purporting to show related ordering on PTFE. I cannot see these small features at all and thus cannot agree with your assertion that “These results strongly support the generality of the findings”. You must either show stronger evidence or delete this paragraph and related assertions.

We agree with the reviewer that the images are not as convincing as for HOPG. Since these experiments on PTFE are not fundamental to the message of the paper, we have decided to follow the reviewer’s suggestion and remove Fig S13 and the related statements in the manuscript.

Page 9, lines 5-6. Perhaps I am missing something but I do not see how you link Fig 1b to all the different arrangements seen in Fig 3a. Why would ribbons – the lowest energy structure in your simulations – not also be dominant in Fig 1b as well? Or to put it another way, how does your simulation fold in the fact that Fig. 1b is for low X_{MeOH} data?

MD simulations show that – even for the disordered model – the surface concentration of MeOH does not directly reflect the bulk concentration. In the presence of ordered phases, the link between the bulk concentration and the composition of the surface layer could differ even more strongly. Fig. 3a only demonstrates that the MeOH/water surface layer exhibits a strong potential for polymorphism, that is reflected in the observation of regions with a different ordered structure. We cannot however, at present, establish a direct link between one of the structures obtained from modeling and the alternative patterns seen in Fig 1b.

It seems plausible to us that the kinetics of formation of the different structures is key to answering this question. But, before exposing our current explanation for the process, we would like to stress that since simulations cannot capture the formation of the different structures but only their relative stability (or free energy), any discussion involving kinetics is bound to be partly speculative. For the sake of clarity, we hereafter call ‘2D assemblies’ structures similar to those presented in Fig. 3a, by opposition to ‘row-like structures’ which refer to ribbon-based 2D assemblies. Experimentally, we observe that working at lower MeOH concentration tends to favour the formation of 2D assemblies, but their nucleation is typically slower than for

the row structures appearing at higher MeOH concentration, and higher temperatures are required. Additionally, the structures do not cover the whole interface and large regions remain disordered and glassy (purple arrow in Fig. 1d). This is in contrast to the self-assembly at higher MeOH concentration that tends to cover the whole interface once initiated, with an assembly front advancing rapidly (Fig. 4a-d). Taken together, these observations suggest that at lower MeOH concentration, the structure nucleating is not necessarily the equilibrium structures which require a 1:1 MeOH/water ratio, but rather 2D structures favoured by the low MeOH/water ratio. The fact that higher temperatures are required also suggest a higher nucleation barrier, presumably due to the low MeOH content. After some time, the interface exhibits many nucleated domains (Fig. 4f) but these fail to grow and cover the whole surface. This could be explained by the large number of 2D structures appearing even on a 10 nm scale and that prevent a coherent growing front, as would be the case for the row-like structures. As a result, the 2D domains appear like randomly branched islands in the glassy layer, where the boundary between the structured 2D domain and the unstructured surrounding may not be able to propagate further having reached an unfavourable local assembly. The assembled 2D domains are however still stable enough (kinetic trap) to prevent re-organisation into a more stable row-like assembly.

An extensive exploration of the X_{MeOH} /Temperature/timescale parameter space is still needed to provide a definitive answer, a task already underway but that goes beyond the message of the present paper. We have added to the following text in the discussion to convey the main points of the above explanation:

Working at lower MeOH concentration forms different patterns with a high degree of heterogeneity at the nanoscale (Fig. S2e-f), and that do not cover the whole interface. Higher nucleation temperatures are also required, suggesting the low alcohol concentration to favour assemblies corresponding to local energy minima (Fig. 3a) rather than the more stable row-like structure. The high degree of polymorphism in the structural molecular arrangement could also explain the difficulty for nucleated domains to grow.

Minor points

1) Page 2, refs 5-6 with water within graphene is probably arising from ions/crystallization within the water – see follow up papers on this topic.

We thank the referee for pointing this out. Referee 1 also suggested using some references by the group of Kay since they may be more relevant. We have now updated this text and moved it to the discussion (as requested by Referee 4).

2) Page 2, lines 7-11. Each of these 3 sentences are rather sweeping statements – you need to back up each with some references.

The sentences have been rephrased to tone down the sweeping statements and wherever suitable references have been added.

3) Page 3, line 7. You say “neither pure alcohol ...” but I assume you do not mean so general and refer just to small chain alcohols like methanol and EtOH. Can you clarify in the text.

The reviewer is indeed correct and the text has been changed to specify MeOH and EtOH since larger alcohols can form monolayers.

4) Figure 1. Can you state the temperatures of this specific example. Can you state the concentration for Fig (a).

This has been added to the caption of the updated figure

5) Figure 4. Can you state what Xmeoh is. Also, on line 8 you say “such as those in fig 4”. Is this a mistake as I cannot seem to see what relevance to Fig. 4.

The reviewer identified a mistake. The caption text was meant to refer to Fig 2d.

6) Similarly, page 11 line 12, where is “Fig. 5e” ??

This is a mistake that has now been corrected (a reference to Figure 4e was intended)

7) Page 12, line 21, do you mean “The low mixing” not “The low the mixing”? or similar? Please clarify.

The reviewer identified a typo that has now been corrected.

8) Just a note – no comment needed. I think the temperature rise from the laser impinging the HOPG will be rather small so would be surprised if this induces nucleation. Do you have clear evidence for this?

We do not have a strict proof of it, but we found that although nucleation always happened with the large laser, it was very difficult to predict or indeed reproduce the conditions in which this happened. When not heating, it was a matter of waiting long enough (sometimes hours) with often an apparent enhancement of nucleation in the region in close vicinity to the tip. When heating, no reproducible nucleation temperature could be determined. Using the small spot laser allowed for much better reproducibility and nucleation could be seen to occur at the expected temperature within minutes. This is of course no proof, but suggests to us that the red laser may indeed play a role.

In Supplementary information:

1) Page 7, Fig S3. Very hard to see any HOPG lattice in (a). Can you use a higher resolution picture or an insert to show.

An insert has been added to the figure in order to better show the HOPG lattice.

2) Page 15, line 2. Do you mean “used in Fig. S11”, not “Fig. 5” ? A bit confusing.

3) Page 17, Fig S14. State what the Temperature is.

We thanks this reviewer for his/her careful appraisal of our manuscript

Reviewer #4 (Remarks to the Author):

The authors investigated the formation of 2-dimensional structures on graphite surface when a solution of water and alcohol is deposited on it. Using AFM they observe the formation of row- structure domains on the surface. The authors used multiscale molecular dynamics simulations to interpret the AFM image.

The resolution of the AFM images is quite poor. A row-structure can be observed but

it is not possible to identify the 2 molecular species in the images. Intermolecular bonds can not be experimentally determined. The calculations are then very speculative: many stable arrangements are possible. Overall the interpretation of the authors is very speculative because the comparison between experimental data and simulation is not convincing.

We agree with the reviewer that the AFM resolution is somewhat limited, if compared with STM or AFM studies in UHV. However, in the present case we are dealing with a soft layer (that can be scratched off by the AFM tip, Fig. S3) imaged in solution at temperatures up to 50C. STM imaging is not easily achievable in these conditions and the resolution of our images is absolutely state-of-the-art and indeed comparable with the best published AFM results in liquid. We nonetheless agree that the images do not allow resolving individual molecules and the proposed explanation and link with the simulation depicts a plausible arrangement rather than a strict proof. This point was also raised by Reviewer 3 and we have taken steps to improve the resolution and interpretation of our AFM images, and ensure their reproducibility. This is visible in an updated Fig. 1 of the manuscript and in the updated Fig. S2.

Despite these issues, we would like to point out that the main message of the paper is the ability for water and short alcohols to form H-bond self-assembled structures at the surface of graphite in solution, in a thermally nucleated process. We feel that this finding is supported by our data, even if the precise molecular arrangement is not fully confirmed. We believe that this result is significant enough to be of interest to the broad readership of Nature Communications because it is completely unexpected and has implications in a wide range of processes where water-alcohol mixtures are used, from fuel cells, to nanofluidics, and self-assembly. The novelty comes from the fact that the interactions rendering this water-alcohol self-assembly possible are not the usual hydrophobic/van der Waals interactions between the alcohol alkyl tail and the graphite substrate as common for longer alcohols (see e.g. Messé et al, J. Coll. Inter. Sci 2003). If this was the case, no self-assembly could take place because of the small size of the methanol/ethanol alkyl chains and the relatively high temperatures. Another type of interactions must be at play, and the simulations suggest that it is a collective effect of the hydrogen bonding network developing between the alcohol and the water. The simulations, although not directly confirmed by the AFM results, are compatible with them and very plausible given the unusual self-assemblies observe experimentally.

For these reasons I can not support this article for publication. It should be submitted to a more specialised journal like J. Phys Chem.

We understand the point of view of the reviewer, but we respectfully disagree for the reasons exposed in our answer to his/her previous question.

In addition the paper is not easy to read because its style is quite confusing. The introduction is for example already discussing the results before they are presented. The structure of the paper should be strongly modified.

We appreciate the reviewer's reservations regarding the style of the paper. It is increasingly common for articles to conclude the introduction with a paragraph briefly

outlining the main results so as to prepare the reader. This might have lead to some confusion and we have therefore rephrased the relevant paragraph of the introduction and moved most of the statements to the discussion section.

Please find below our point-by-point answer to the comments of Reviewers 1 and 3 who made useful suggestions to improve our manuscript.

For the sake of clarity, the comments from the Reviewers have been embedded in the text and appear in black our answer to the comments in shown in blue, and any citation from the text of the manuscript is shown in red.

A list of all the changes made to the paper as a result of the Reviewer's comments is given after our detailed responses to the Reviewers

We are confident that this revised version of the manuscript fully addresses the considerations of the Reviewers and we hope that you will find it suitable for publication.

Best regards,

Kislon Voitchovsky (on behalf of all the authors)

Answer to Reviewer #1

The authors have improved the manuscript, but a few minor questions remain and changes made to the manuscript raise a few new questions. I suggest that the following comments are considered before publication.

1. Lines 21-22: "Here we show, that mixtures of water with methanol or ethanol can form a novel type of interfaces with hydrophobic solids."

The abstract gives the impression that binary water-ethanol mixtures may also nucleate new structures at high temperatures. This does not appear to be covered in the manuscript and the sentence needs to be rephrased.

We have removed mention of the ethanol spiking from the abstract, as it could indeed be misconstrued as referring to binary water-ethanol mixtures.

2. Lines 63-67: "The structures remain stable at room temperature, pointing to the collective effect of an extended H-bond network. This is significant because most layering of alcohol molecules at interfaces is based on van der Waals interactions^{18,19} which are not strong enough to stabilise short alcohols such as methanol or ethanol at the interface with immersed solids at room temperature."

What do you mean by "stabilize" in the last sentence?

The strength of the molecule-surface interactions is similar for the different systems treated here; the question is rather if a specific system will form solid- or liquid-like structures at the interface. Also, polar OH groups play an important role for the intermolecular interactions between small alcohol molecules, making the first part of the sentence unclear in relation to the H-bond network mentioned in the preceding sentence.

We concur with the Reviewer that the sentence is slightly ambiguous. We have modified it to clarify our reasoning, namely that van der Waals forces between molecules are dominant in the case of longer aliphatic chains, but that they cannot explain the behavior of methanol or ethanol for which polar and H-bond interactions are dominant. The modified sentence reads

as follow:

This is significant because layering of long alcohol molecules at interfaces is usually based on van der Waals interactions^{18,19} which are not strong enough to drive the self-assembly of short alcohols such as methanol or ethanol at the interface with immersed solids at room temperature.

3. Lines 73-77 would fit better in a concluding paragraph, or may be removed. This paragraph has now been moved to the conclusion paragraph of the paper.

4. Line 109: "... 1:1 water-methanol mixture spiked with a small amount of ethanol." Comments are required about how much ethanol that was actually added to the ternary solution, and why this plays a role for the observed resolution. We used the term 'spiked' because the amount of ethanol added to the 1:1 water-methanol solution is small (< 1%) but it is difficult to provide an experimentally precise value due to possible evaporation between the adjunction of ethanol and the acquisition of the image. We have now added "<1%" in the text. Regarding the improved resolution we can only speculate on the precise role of the ethanol. We have added the following sentence: The clearer resolution with EtOH may be due to the slightly larger periodicity making the difference close to the resolution limit for this system.

5. Lines 176-178: "However, this does not necessarily signal a deficiency of the model, but rather provides an indication of the long time scale involved in the dynamics of this system, and of the activated nature of the process by which the experimentally observed 2D-pattern is formed."

Slow dynamics do not necessarily say anything about the activated nature of the process and the sentence needs to be rephrased.

We agree that slow dynamics is not a sufficient condition to prove the activated nature of a process. Rather, it is the activated behaviour observed experimentally that is consistent with the behaviour seen in the simulations. We have modified the sentence to clarify the causal relation:

However, this does not necessarily signal a deficiency of the model, but rather provides an indication of the long time scale involved in the dynamics of this system and is consistent with the activated nature of the process by which the experimentally observed 2D-pattern is formed.

6. Lines 317-331: It is worth pointing out that an alcohol monolayer at the vacuum-graphite interface often has a substantially higher melting point than the corresponding bulk melting point: ethanol 214 K (bulk 156 K), propanol 230 K (147 K), butanol 255 K (184 K); see Moreshige and coworkers in publications 1990-1999. Methanol is actually an exception with a monolayer melting point of 143 K (bulk 175 K). In addition, it may be worth mentioning that in the water-graphite system investigated here, methanol tends to concentrate at the interface due strong water-water interactions in the bulk and a stronger methanol-graphite interaction compared to water-graphite. The interfacial methanol may therefore be considered as confined between the liquid and solid phase, which obviously results in new solid-like structures and a further increase of the melting temperature.

We thank the reviewer for point this out and agree that the 'confinement' of methanol could in

principle explain the results. However, we feel that it is one of the possible ways to interpret the thermodynamic drive towards self-assembly, especially considering the fact that our results indicate that the final structures to incorporate both water and methanol in an alternated fashion (i.e. not a pure methanol structure). We have added the following sentences to the discussion:

We note that in vacuum, alcohol monolayers on graphite tend to melt at higher temperatures than in liquid, for a example around 260 °K for pentanol²⁴. A ‘confinement’ of interfacial MeOH between the graphite and the bulk water could, in principle, explain the observed structures and increased melting temperature. However, since both water and MeOH are present in the structures, this interpretation would be limited to the thermodynamics driving the self-assembly, not the resulting structures.

[24. Morishige, K. & Sakamoto, Y. Melting of n-butanol and n-pentanol monolayers adsorbed on graphite: Effect of molecular length on melting. *J. Chem. Phys.* **103**, 2354–2360 (1995).]

7. Lines 369-371: “Here neither water nor the alcohol molecules experience an interaction sufficiently strong with the graphite surface to remain stably at the interface at room temperature, but a collective effect is necessary.”

The strength of the molecule-surface interactions is similar for the different systems treated here; see comment No 2 above.

We have reworded the sentence as follow:

Here neither water nor the alcohol molecules remains stably at the interface between the liquid and graphite at room temperature, but a collective effect is necessary.

8. Some linguistic issues previously listed by Reviewer No. 2 are not yet corrected.

We verified all the issues previously reported by Reviewer 2 and addressed them. We have also carefully proofread the manuscript.

We thank the Reviewer for his/her detailed comments, which allowed us to improve our manuscript.

Reviewer #3 (Remarks to the Author):

The re-submission is now acceptable for publication.

there is a small correction in the caption of Figure 1.

* At low MeOH concentration (b) other..... I think you mean (d) here.

The reviewer is indeed correct. We thank him/her for their critical reading of our manuscript and for spotting this important typo that has now been corrected.

List of the changes

The changes refer to lines in the manuscript version **NCOMMS-16-04615A**

1. Lines 21-22: Improved abstract to avoid ambiguity on the content of our study
2. Lines 63-67: Clarification of a statement regarding the interaction between the molecules and the graphite
3. Lines 73-77: Concluding sentence of the introduction moved to the conclusion
4. Lines 109-... (caption figure 1): Clarifications added regarding the amount of ethanol used to spike the solution and the resulting effect on the AFM imaging resolution.
5. Lines 120 (caption figure 1): A typo referencing different parts of the figure has been corrected
6. Lines 176-178: Rewording of a sentence to avoid ambiguity
7. Lines 317-331: Addition of a small paragraph and a reference regarding the formation and stability of alcohol monolayers on graphite in vacuum.
8. Lines 369-371: Rewording of a concluding sentence to avoid some ambiguity regarding the interaction between the molecules and the graphite surface.